# Clove Oil Delays Rather Than Prevents Scur/Horn Growth in Dairy Cattle

**DOI:** 10.3390/vetsci6040102

**Published:** 2019-12-13

**Authors:** Mhairi Sutherland, Alan Julian, Frances Huddart

**Affiliations:** 1AgResearch Ltd., Ruakura Research Centre, Hamilton 3214, New Zealand; frankie.huddart@agresearch.co.nz; 2New Zealand Veterinary Pathology/IDEXX Laboratories, 20 Maui Street, Hamilton 3200, New Zealand; Alan-Julian@idexx.com

**Keywords:** calf, cautery, disbudding, cryoablation, horns, scurs

## Abstract

The objective of this study was to evaluate if administration of clove oil prevents scur/horn growth in dairy cattle long term. At approximately 4 days of age, calves had one of four treatments assigned to each horn bud: (1) clove oil administered subcutaneously under the horn bud (CLOV, *n* = 132); (2) cautery disbudded and the horn bud removed (BUDOFF, *n* = 126); (3) cautery disbudded and the horn bud tissue left intact (BUDON, *n* = 129); (4) a liquid nitrogen filled probe applied to the horn bud area (CRYO, *n* = 131). At approximately 16 months of age, all cattle were checked for scur or horn development. A sub-set of scurs/horns from the CLOV cattle were removed to evaluate tissue and structural development. In total, 5% of CLOV buds developed into horns and 63% into scurs; 10% of the scurs looked like normally developed horns but they were not attached to the skull. Cautery disbudding prevented scur and horn development in cattle when the horn bud tissue was removed, but some scur growth was observed in the BUDON treatment. CRYO was 100% ineffective at preventing scur/horn growth. Injecting clove oil under the horn bud appeared to delay horn development, but not prevent it, when administered to 4 day old dairy calves.

## 1. Introduction

Disbudding, a procedure commonly conducted on dairy calves to prevent horn growth, is often performed using a heated cautery iron [1,2,3]. Disbudding is performed to prevent horned cattle causing injuries to people and other animals, especially during transport [2]. Cautery disbudding destroys the horn bud tissue, but if performed without pain relief, causes pain as indicated by changes in calf behaviour and physiology (reviewed by Stafford and Mellor [2]). A multimodal approach to pain management involving administration of local anaesthesia in conjunction with a non-steroid anti-inflammatory drug can mitigate the pain associated with the initial nociceptive damage caused by the cautery iron and the longer-term inflammatory pain associated with thermal burns and has been demonstrated in calves to reduce or abolish the pain response to disbudding [4,5,6,7]. However, even if pain relief is provided, disbudding still causes tissue damage that can lead to infection and longer-term pain [8,9]. In addition, providing pain relief has economic considerations such as cost of the drugs, veterinarian and extra labour. Therefore, there remains a need to evaluate alternatives to cautery disbudding that are practical and improve calf welfare.

One alternative is to select for polled dairy cattle which would ultimately negate the need to disbud calves; however, genetic companies first need to make greater advancements in high quality polled sires to encourage adoption of this practice by industry [10]. Two novel methods of disbudding calves have been described in the literature—clove oil injected into the horn bud region [9,11,12] and cryosurgery [13,14]. At high concentrations, eugenol the active ingredient of clove oil, can cause cellular necrosis [15,16]. Clove oil has been shown to prevent horn growth in calves [11] and goats [17] when injected into the horn bud region and this is likely due to its cellular necrotic properties. Moreover, clove oil has also been shown to have analgesic [18], anti-oxidative and anti-inflammatory properties (refer to review by Kamatou et al. [19]), and this may explain why clove oil injected under the horn bud appeared to cause calves less initial pain and tissue damage than cautery disbudding [12]. Cryosurgery is the process of applying extreme cold temperatures to cause localised cell destruction through crystallization of water within the cells and has been used for the treatment of cutaneous lesions and tumours in human medicine [20]. Cryosurgery does not appear to cause less pain than cautery disbudding, but it does cause less external tissue damage [14], so in conjunction with pain relief could be a possible alternative to cautery disbudding. In an initial study, cryosurgery applied for 10 s or 15 s prevented horn growth in 19% and 33% of calves respectively, hence applying liquid nitrogen for longer could potentially improve its efficacy [14]. As previous studies suggest that clove oil and cryosurgery could potentially provide alternatives to cautery disbudding, further evaluation of the efficacy of these methods to prevent scur/horn growth is needed. Results from a previous study conducted by our group, designed to evaluate the efficacy of clove oil, cryosurgery and cautery disbudding (horn bud tissue removed or left intact) to prevent horn growth, reported that these methods were 87%, 2%, 100% and 91% successful at preventing horn and scur development by 6 months of age, respectively [9]. However, the long-term effect of these treatments on efficacy to prevent horn growth is unknown. Therefore, the objective of this study was to evaluate the effect of clove oil (injected under the horn bud), cautery disbudding and cryosurgery on scur/horn growth development in dairy calves up to 16 months of age.

## 2. Materials and Methods

All procedures involving animals were approved by the Ruakura Animal Ethics Committee (Protocol No. 14215) under the New Zealand Animal Welfare Act 1999.

At approximately 4 days of age (4.1 ± 0.94 d of age, mean ± SD), 271 dairy heifer calves were enrolled on three dairy farms located in the Waikato region, North Island, New Zealand. Each calf had one of four treatments randomly assigned to each horn bud: (1) clove oil (CLOV, *n* = 138 buds); (2) cautery disbudded with the horn bud tissue removed (BUDOFF, *n* = 134 buds); (3) cautery disbudded with the horn bud tissue left intact (BUDON, *n* = 135 buds); (4) cryosurgery (CRYO, *n* = 135 buds).

Application of treatments are described in detail in Sutherland et al. [9]. Briefly, calves were manually restrained and the hair within the horn bud area was shaved. All calves received a cornual nerve block using 4 mL of local anaesthetic (2% lignocaine hydrochloride; Lopain, Ethical Agents Ltd., Auckland, New Zealand) and a non-steroidal anti-inflammatory drug (Loxicom^®^, Norbrook^®^ Laboratories Limited, Auckland, New Zealand) administered subcutaneously. At least 5 min after administration of the local anaesthetic, the calf’s head was restrained, and the allocated disbudding treatment was performed to each horn bud: cautery disbudding involved placing an electric cautery iron (“Quality” electric debudder, 230 V, 190 W; Lister GmbH, Lüdenscheid, Germany) on the horn bud for approximately 15 s and the horn bud was then either left intact (BUDON) or removed (BUDOFF); cryosurgery involved placing a flat cylindrical hollow probe, cooled by filling the probe with liquid nitrogen, onto the horn bud for 30 s using a commercial spray applicator (Cry-Ac, Bry-Mill Cryogenic Systems, Ellington, CT, USA); clove oil (0.5 mL, C8392, 83–85% eugenol, Sigma-Aldrich, Saint Louis, MO, USA) was injected subcutaneously under the centre of the horn bud. Calves were then returned to the home pen and reared per normal farm practice for each farm.

At approximately 16 months of age, calves were assessed for signs of scur or horn growth by a veterinarian and trained technician. Assessors were blind to the disbudding treatments. Assessors classified horn growth visually and determined the difference between horns and scurs: Horn: a growth that looked like a normally developed horn and was attached to the skull; Scur: a normally developed horn but NOT attached to the skull OR an abnormally developed horn NOT attached to the skull. The disbudding procedure was considered successful if there was no evidence of horn or scur growth.

After the 16 months assessment period, scurs and horns were removed by a veterinarian to prevent injury to other animals or animal handlers. Prior to dehorning, all animals received a cornual nerve block and a subcutaneous ring block (Lopaine 2% lignocaine, Ethical Agents Veterinary Marketing Ltd., Auckland, New Zealand) and a non-steroidal anti-inflammatory drug (Loxicom^®^, Norbrook^®^ Laboratories Limited, Auckland, New Zealand). A subset of scurs (*n* = 21; *n* = 20 CLOV, *n* = 1 CRYO) and one untreated horn were cross-sectioned, photographed and prepared for histological analysis by a veterinary pathologist.

The horns and scurs were fixed in 10% buffered formalin. After fixation they were bisected and a 5 mm wide slice was trimmed from the midline. If there was bone in the sample, the slice of tissue was decalcified in 10% formic acid until softened. The horn was then softened using 10% Potassium hydroxide (KOH). The tissues were embedded in paraffin wax for histological examination. Histologic sections were cut at 5 μ, stained with haematoxylin and eosin, and examined for pathologic changes using light microscopy.

Data are presented descriptively as means.

## 3. Results

Efficacy of the different treatments to prevent scur/horn growth 6 months after disbudding are reported in Sutherland et al. [9] and Table 1. The authors did not anticipate further scur/horn development after 6 months, but calves were reassessed at 16 months as part of the ethics protocol. At 16 months of age, 12 calves had either died or been sold (six buds treated with CLOV, eight with BUDOFF, six with BUDON, four with CRYO). No deaths were associated with any of the disbudding treatments. Final numbers of horn buds assessed at 16 months for each treatment were: CLOV (*n* = 132 buds), BUDOFF (*n* = 126 buds), BUDON (*n* = 129 buds) and CRYO (*n* = 131 buds).

The clove oil treatment resulted in 5% of buds developing into horns and 63% into scurs (Table 1). Of these 83 scurs, eight looked very similar to a normally developed horn (Figure 1); however, they were defined as a scur as they were moveable and not attached to the skull.

The cross-section of a normally developed, non-disbudded (untreated) horn consisted of an outer layer of cornual epidermis, followed by a dermal layer, a fibrous tissue layer and a cornual diverticulum (Figure 2a). The internal characteristics of the scurs from CLOV treated buds, that looked like a normally developed horn but were not attached to the skull, were similar to the normal horn but did not have a cornual diverticulum (Figure 2b) and this was also similar for scurs from CLOV treated buds that looked like small deformed horns (Figure 2c).

BUDOFF and BUDON were the most efficacious treatments, with BUDOFF being the most efficacious as this procedure resulted in no scur or horn development over the entire 16 month period (Table 1). In total, 8% of the buds that received the BUDON treatment developed into scurs.

Over the entire 16 month period, CRYO was the least effective treatment with 76% buds developing into horns and 24% into scurs (Table 1). In total, 99% of scur/horn development occurred within the first 6 months, with only one bud developing into a scur as a result of the original treatment by the 16 month observation period.

## 4. Discussion

By 16 months of age, 5% of buds treated with clove oil developed into horns and 63% into scurs. Cautery disbudding was the most efficacious method of preventing scur/horn growth, with the BUDOFF technique resulting in no scur or horn growth; however, leaving the bud tissue intact after cautery disbudding did result in some scur/horn development. Using the methodology described in this study, clove oil and cryosurgery resulted in 68% and 100% of buds developing into scurs or horns respectively, which is not acceptable for on-farm management of dairy cattle.

Injecting clove oil under the horn bud of 4 day old dairy calves only prevented scur/horn development in 32% of the treated horn buds in the present study. However, Molaei et al. [11] reported that injecting clove oil into the horn bud region of 5 day old calves prevented horn growth in 100% of treated animals (*n* = 4 horn buds). Similarly, Molaei et al. [17] and Abbasi et al. [21] found that injecting clove oil into the horn bud region of 5 day old goat kids prevented horn growth, but this was also only evaluated in a small number of animals. Conversely, Hempstead et al. [22] found that injecting clove oil into the horn bud of 4 day old goat kids resulted in a 72% and 21% probability of scurs and horns developing, respectively. Differences in the success of clove oil to prevent scur/horn growth among studies could be due to several factors, including animal breed, type of clove oil injected (e.g., source of clove oil/eugenol, concentration of eugenol), dose/volume of clove oil injected and how it was administered. Therefore, it would be of interest to determine if cattle breed or method of administration (e.g., age of calf at time of administration, dosage/volume or source of clove oil) would improve the efficacy of clove oil to prevent scur/horn growth in cattle. However, if the method of clove oil administration is refined so as to efficaciously prevent scur/horn growth there would still be several barriers that would need to be overcome before it would likely be adopted by industry, such as the age of disbudding (e.g., 4 days vs. 2 weeks), method of administration, potential drug residues and drug regulatory issues.

At 16 months of age, 68% of horn buds injected with clove oil had developed into scurs or horns (results from the present study); however, at 6 months of age only 13% of horn buds injected with clove oil had developed into scurs and none into horns [9]. Therefore, it appears as though clove oil may delay rather than prevent horn growth. Another interesting finding from this study was that several scurs (10%) that developed from horn buds injected with clove oil looked very similar to a normally developed horn (Figure 1); however, they were defined as a scur as they were moveable and not attached to the skull. The development of a normal ruminant horn involves keratinisation of the horn bud epidermis and ossification of the underlying dermis and hypodermis [23], the cornual diverticulum of the frontal sinus then forms a cavity within the horn by about 6 months of age [24]. If the horn bud epidermis is not completely destroyed, then keratinisation of some epidermal cells can occur resulting in horny growths (scurs) that are not attached to the skull and generally look like small deformed horns. In the present study, we were expecting the cellular necrotic properties of eugenol to destroy the entire horn bud tissue and prevent any scur or horn development; however, injecting clove oil under the horn bud may have resulted in necrosis of only the horn bud cells under the bud. This process may have prevented normal attachment to the skull and formation of the cornual diverticulum, whereas the cells above the bud may have remained unaffected and continued to proliferate and develop into a normal looking horn. Moreover, when administering clove oil is was often difficult to restrain the calves’ head completely, which could also have led to inadequate placement of the oil. It is unclear from the present study what mechanism was involved in causing the delay in scur/horn development in clove oil treated buds. It would be of interest to know if the scurs that looked like normally developed horns would have eventually developed a cornual diverticulum; however, for the welfare of the animals we wanted to remove these scurs before they got bigger. Overall, based on the results from the present study, injecting clove oil using the described methodology it is not recommended as an alternative method to cautery disbudding to prevent horn growth in dairy calves.

The removal of the cauterised horn bud tissue after cautery disbudding in calves is performed to prevent horn growth due to remaining viable germinal epithelial cells. The results from the present study support this as being the most efficacious method to prevent horn and scur development in dairy calves. Several operators prefer to leave the horn bud tissue intact after cautery disbudding for different reasons; however, this method resulted in the development of scurs (7% of buds evaluated at 16 months of age) and also an increased incidence of infection within the first month after disbudding [9]. It would be of interest to evaluate if leaving the horn bud tissue intact after cautery disbudding has any other welfare implications, for example does the extra pressure applied to the calves’ head as a result removing the horn bud tissue cause more discomfort to the calves. In addition, calves in the present study were cautery disbudded at approximately 4 days of age whereas it is common practice to cautery disbud calves at 4–6 weeks of age [2]; it would be of interest to evaluate if age effects efficacy of cautery disbudding to prevent scur/horn growth.

Cryosurgery was the least effective method of preventing horn growth in the present study. This may be due to the method of application; Sutherland et al. [14] reported that spraying liquid nitrogen directly onto the horn bud site had a 19% and 33% success rate at preventing horn and scur growth when applied for 15 or 10 s respectively. Similarly, Bengtsson et al. [13] found that applying liquid nitrogen using a cryoprobe was less efficacious at preventing horn development in calves than applying liquid nitrogen directly onto the skin. Therefore, it does not appear that the method of cryosurgery used in the present study caused sufficient destruction of the horn bud cells to prevent scur/horn development. Studies described in the human medicine literature indicate that tissues need to get to between −30 °C and −50 °C to cause cell death (depending on the cell type), also a repetition of a freeze-thaw cycle is often used to enhance the destructive properties of cryosurgery [20]. Further research investigating the optimum temperature and the use of a freeze/thaw cycle to cause cell destruction in horn bud tissue is needed. However, conducting cryosurgery on-farm has several practical implications, such as managing liquid nitrogen on farm and worker safety, which would need to be resolved before this method of disbudding could become a practical alternative to cautery disbudding. The results from the present study; however, indicate that cryosurgery applied using the methodology described in the present study cannot be recommended as an alternative method to cautery disbudding to prevent horn development in dairy cattle.

## 5. Conclusions

Cautery disbudding was the most effective of the methods evaluated to prevent scur/horn growth in dairy calves, regardless of whether the horn bud tissue was removed or left intact, and liquid nitrogen applied for 30 s using a probe was ineffective. Though injecting clove oil under the horn bud of 4 day old dairy cattle initially appeared to prevent scur/horn growth in the majority of animals treated, the results from this study suggest that it only delayed scur/horn development. Therefore, the method of clove oil administration used in the present study cannot be recommended as an alternative to cautery disbudding. There is a need to continue identifying and evaluating possible alternatives to cautery disbudding, such as breeding for polled genetics with greater genetic merit, but until there are alternatives that are practical, cost effective and readily available to farmers, it appears as though cautery disbudding is the most efficacious method of preventing scur/horn growth. However, cautery disbudding causes pain, therefore, until an alternative is developed pain relief should be provided when cautery disbudding calves.

## Figures and Tables

**Figure 1 vetsci-06-00102-f001:**
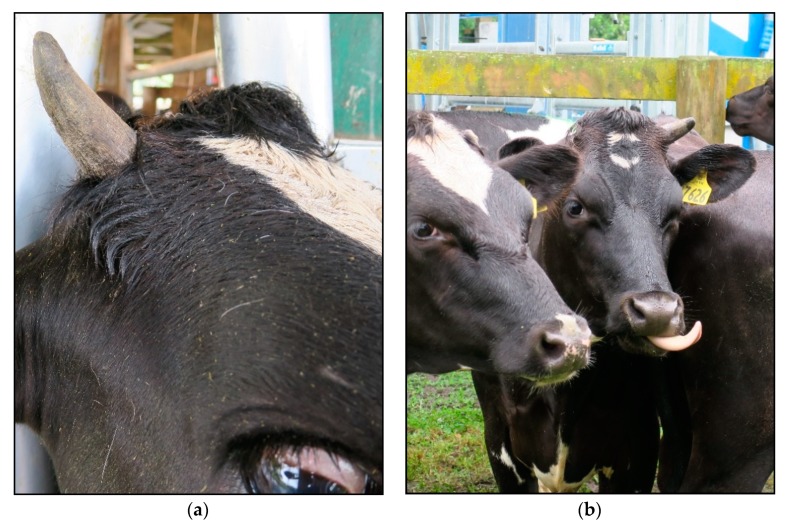
Image (**a**) is a picture of a normally developed horn that was not disbudded and (**b**) is a picture of a scur that developed after clove oil was injected subcutaneously under the horn bud at approximately 4 days of age.

**Figure 2 vetsci-06-00102-f002:**
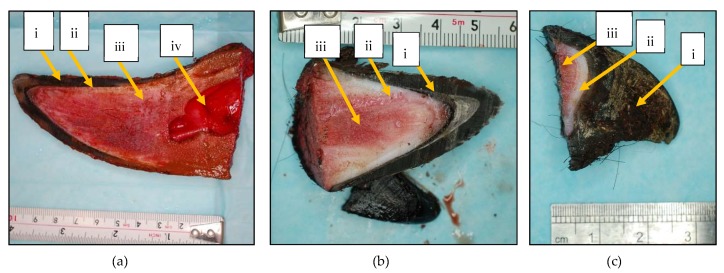
Images of the cross section of a normally developed (untreated/non-disbudded) horn (**a**) and scurs from cattle that had clove oil injected subcutaneously under the horn bud at approximately 4 days of age that looked like a normally developed horn (**b**) and a small distorted horn (**c**). (i) cornified horn: stratum corneum of cornual epidermis, (ii) dermis, (iii) cornual process of the frontal bone, (iv) cornual diverticulum of frontal sinus.

**Table 1 vetsci-06-00102-t001:** Number of buds with no horn growth, scurs or horns at six and 16 months of age after calves were disbudded by injecting clove oil subcutaneously under the horn bud (CLOV), using a cautery iron and removing the horn bud tissue (BUDOFF) or leaving the bud tissue intact (BUDON), and applying a liquid nitrogen cooled probe applied to the horn bud area (CRYO).

	6 Months ^1^				16 Months			
Treatment	No. of Buds	No Growth	Scurs	Horns	No. of Buds	No Growth	Scurs	Horns
CLOV	135	118	17	0	132	43	83	6
CAUT (BUDOFF)	130	130	0	0	126	126	0	0
CAUT (BUDON)	131	119	11	1	129	119	10	0
CYRO	134	2	30	102	131	0	31	100

^1^ Results derived from Sutherland et al. [9].

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
