# Peer review of "Clove Oil Delays Rather Than Prevents Scur/Horn Growth in Dairy Cattle"

_vetsci, 2019, doi:10.3390/vetsci6040102_

Round 1

Reviewer 1 Report

Overall, I really enjoyed reading this article. It is concise, well written, and addresses a very interesting and novel method for horn removal. The results of this analysis have important implications for dairy producers. Find below a few minor comments and edits that I would like to see addressed before publication.

The introduction is well written, but I think it currently lacks broad motivation for why disbudding is necessary and why it is important to consider alternative methods for removing horns. There is obviously the issue of animal welfare/pain relief which is discussed. There is also the issue of genetic merit (mainly milk production) as it relates to polled genetics which is captured in the Spurlock et al. (2014) reference. Lastly, there are economic motivations. That is, even if we can identify alternatives to cauterization, are they cost-effective? A recent article by Thompson et al. (2017) provides insights into the economic considerations of alternative disbudding methods.

Thompson, N.M., N.O. Widmar, M.M. Schutz, J.B. Cole, and C.A. Wolf. 2017. “Economic considerations of breeding for polled dairy cows versus dehorning in the United States.” Journal of Dairy Science 100:4941-4952. https://doi.org/10.3168/jds.2016-12099.

Line 96 change till to until

Line 102 change first are to after

Line 105 change being to been

Line 225 cautery disbudding was the most effective of the methods evaluated to prevent horn growth…

I also find the Conclusions to be a bit terse. I think the conclusions drawn are certainly supported by the data, but I think the authors could do more to draw out the industry implications of these findings. Mainly, if cauterization remains the most efficacious disbudding method, how does the industry move forward with consumer interest in animal welfare issues? Mandated use of pain relief? Investments in polled genetics?

Author Response

Au: Thank you for reviewing this paper and your very helpful comments. We have replied to each of the comments below.

Overall, I really enjoyed reading this article. It is concise, well written, and addresses a very interesting and novel method for horn removal. The results of this analysis have important implications for dairy producers. Find below a few minor comments and edits that I would like to see addressed before publication.

Au: Thank you

The introduction is well written, but I think it currently lacks broad motivation for why disbudding is necessary and why it is important to consider alternative methods for removing horns. There is obviously the issue of animal welfare/pain relief which is discussed. There is also the issue of genetic merit (mainly milk production) as it relates to polled genetics which is captured in the Spurlock et al. (2014) reference.

Au: Some more information about why disbudding is necessary has been included in the introduction as well as why it is important to consider alternative methods of pain relief.

Lastly, there are economic motivations. That is, even if we can identify alternatives to cauterization, are they cost-effective? A recent article by Thompson et al. (2017) provides insights into the economic considerations of alternative disbudding methods.

Thompson, N.M., N.O. Widmar, M.M. Schutz, J.B. Cole, and C.A. Wolf. 2017. “Economic considerations of breeding for polled dairy cows versus dehorning in the United States.” Journal of Dairy Science 100:4941-4952. https://doi.org/10.3168/jds.2016-12099.

Au: This is an important consideration. Any alternative must be practical to farmers and cost-effective. These points have been included in the introduction and discussion.

Line 96 change till to until

Au: Changed.

Line 102 change first are to after

Au: Changed

Line 105 change being to been

Au: Changed

Line 225 cautery disbudding was the most effective of the methods evaluated to prevent horn growth…

Au: Changed

I also find the Conclusions to be a bit terse. I think the conclusions drawn are certainly supported by the data, but I think the authors could do more to draw out the industry implications of these findings. Mainly, if cauterization remains the most efficacious disbudding method, how does the industry move forward with consumer interest in animal welfare issues? Mandated use of pain relief? Investments in polled genetics?

Au: These are very important points and have now been added to the conclusion.

Reviewer 2 Report

Major Revision:

Conclusion and title are unacceptable due to the results presented in this study. Insufficient success can be caused, for example, by incorrect dosage, wrong technique, age of the animals and many more factors! Weaknesses of the study should be discussed in the discussion. Conclusion can only be made on the basis of results! Results should be presented objective.

What´s the reason for local anaesthesia? Clove oil is locally aesthetically active.

Is there a breed dependency observed?

Is there an age dependency observed? Is there a significant difference between calves treated with an age of 1 and 2 days to calves treated with an age of 3 and 4 days?

Macroscopic investigation of horn might be interesting- but histological examination is much more informative and should be done.

In presented form the manuscript can be recommended for a publication after major revision.

Author Response

Au: Thank you for reviewing this paper and your very helpful comments. We have replied to each of the comments below.

Conclusion and title are unacceptable due to the results presented in this study. Insufficient success can be caused, for example, by incorrect dosage, wrong technique, age of the animals and many more factors! Weaknesses of the study should be discussed in the discussion. Conclusion can only be made on the basis of results! Results should be presented objective.

Au: We agree that insufficient success can be caused by many factors including dosage/volume, technique, age of the animals and many more and some of these factors that can affect the efficacy of clove oil has been discussed in the discussion (L185-189). However, this paper is a presentation of the results found based on the method used in this paper. In that respect, it is unclear why the reviewer does not think that our title, results or conclusion are representative of our results. 132 horn buds/ cattle were injected with clove oil and 68% of those buds developed into scurs/horns. We used a large sample size and cattle on 3 different farms, so I think it is acceptable to say that clove oil did not prevent scur/horn growth in this study based on the methodology used by the authors. Also we found that the growth was delayed as we had a success rate of 87% at 6 months of age but only 32% at 16 months of age, so I think we can safely say that in this study horn growth was delayed. However,  we have tried to make clarifications throughout the paper to make it more clear to the readier that the results are based on the methodology we used.

What´s the reason for local anaesthesia? Clove oil is locally aesthetically active.

Au: We used local anaesthetic to prevent the potential pain caused by injecting clove oil. Though clove oil has been shown to have anaesthetic properties in some species for some procedures, it has not been shown to cause no pain when used to disbud calves (see Sutherland et al 2018- behavioural and physiological responses to clove oil injected under the horn bud of calves. Appl. Anim. Behav. Sci. 204:29-36). As the purpose of this study was not to evaluate the pain caused by the different procedures we decided to be on the safe side and give local just in case clove oil does cause any pain or discomfort. Also, we wanted to be consistent in our handling between treatments and as cautery does cause pain, we decided to give all calves/treatments local.

Is there a breed dependency observed?

Au: No. All calves used in this study were dairy cattle (Friesian, Friesian cross or Jerseys).

Is there an age dependency observed? Is there a significant difference between calves treated with an age of 1 and 2 days to calves treated with an age of 3 and 4 days?

Au: No. We tried to disbud most of our calves as close to 4 days of age as possible, so our calves were either 3,4, or 5 days of age: 63%, 64%, 77% of calves had either scur/horn growth when disbudded at 3, 4 and 5 days of age respectively. However, age is likely an important factor effecting the efficacy of clove oil and has been discussed in the discussion (L185-189).

Macroscopic investigation of horn might be interesting- but histological examination is much more informative and should be done.

Au: Histological examination was conducted and the results are described in L137-142. We found nothing different in the cell types between normal and clove horns/scurs.

Reviewer 3 Report

This study is a longer term follow up of Sutherland et al 2019 where the same treatment cohorts of calves that underwent different alternative disbudding techniques (at the horn bud level) were followed to 16 months of age to determine horn and scur regrowth. Results of this study show that cryo was inefficient in preventing scur/horn growth but only to 6 months with no new growth to 16. CLOV prevented scur/horn growth at 6 months, but a large proportion of calves had scurs at 16 months.

Overall, interesting and relevant addition to the literature, especially as it relates to alternative methods of disbudding and their effectiveness. Please consider the following suggestions and additions, especially as it relates to clarifying the results of the study (vs. Sutherland et al 2019) for the reader:

L37: Please add one or two sentences for both clove oil and cryoablation on mechanasim of action. Please add some discussion to discussion section related to mechanism of action as to why CLOV and CRYO may not be a good choice for destroying corneal germinal epithelium.

L49: This sentence does not correspond to results published in Sutherland et al 2019. Please reword to state clearly that BUDDOFF = 100%; BUDON = 91%; CRYO = 1%; CLOV = 87%, ie:

Results from a previous study conducted by our group, designed to evaluate the efficacy of clove oil, cryoablation and cautery disbudding (horn bud tissue removed or left intact) to prevent horn growth, found that these methods were 87%, 1%, 100% and 91% successful at preventing horn and scur development by 6 months of age, respectively [9].

L81: To clarify, assessors classified visually and determined the difference between horns and scurs as such:

               Horn: looked like horn and was attached to the skull

Scur: normally developed horn but NOT attached to the skull OR abnormally developed horn NOT attached to skull

Is there a published “veterinary (or otherwise)” definition of horn or scur that you can reference (other than a

histological one?)

L87: The 6 month results were reported in Sutherland et al 2019? If so, please limit your results and discussion to just the 16 month results. Otherwise, it is confusing for the reader which results are being discussed. Conversely, add a table with results (treatment, scur, horn development (proportions)) at 6 and 16 months, similar to Table 1 from Sutherland et al 2019

L91: What treatment group were these scurs/horn from? All from the CLOV group?

L112: Seems like the scurs that developed from 6 to 16 months associated with CRYO were because of incomplete scur/horn removal at 6 months, and not a result of the initial CRYO treatment. Please clarify and/or discuss this for the reader.

L134 (figure 1): Image (a) is a picture of a heifer with a horn that was not disbudded? And the point of the figure is to show that figures a and b look very similar?

L177: Please be clear and consistent when discussing results whether referring to 6 or 16 month results.

L194: Please provide reference for this statement: it is common practice (in what country) to cautery disbud calves at 5w of age.

L201: Please compare your results (related to CLOV) to those reported elsewhere in the literature, and discuss discrepancies and/or why you think those discrepancies may exist.

L224: Conclusions: Please be consistent with use of horn vs scur.

Author Response

Au: Thank you for reviewing this paper and your very helpful comments. We have replied to each of the comments below.

L37: Please add one or two sentences for both clove oil and cryoablation on mechanasim of action. Please add some discussion to discussion section related to mechanism of action as to why CLOV and CRYO may not be a good choice for destroying corneal germinal epithelium.

Au: This information has been added to the introduction and the discussion.

L49: This sentence does not correspond to results published in Sutherland et al 2019. Please reword to state clearly that BUDDOFF = 100%; BUDON = 91%; CRYO = 1%; CLOV = 87%, ie:

Results from a previous study conducted by our group, designed to evaluate the efficacy of clove oil, cryoablation and cautery disbudding (horn bud tissue removed or left intact) to prevent horn growth, found that these methods were 87%, 1%, 100% and 91% successful at preventing horn and scur development by 6 months of age, respectively [9].

Au: The reporting of the results has been changed as suggested above.

L81: To clarify, assessors classified visually and determined the difference between horns and scurs as such:

Horn: looked like horn and was attached to the skull

Scur: normally developed horn but NOT attached to the skull OR abnormally developed horn NOT attached to skull

Au: This section has been changed based on the reviewer’s comments.

Is there a published “veterinary (or otherwise)” definition of horn or scur that you can reference (other than a histological one?)

Au: Not that we are aware of.

L87: The 6 month results were reported in Sutherland et al 2019? If so, please limit your results and discussion to just the 16 month results. Otherwise, it is confusing for the reader which results are being discussed. Conversely, add a table with results (treatment, scur, horn development (proportions)) at 6 and 16 months, similar to Table 1 from Sutherland et al 2019

Au: Agreed. Reference to 6 months results has been removed from the results section. Also, a table has been included that has both the 6 and 16 month results.

L91: What treatment group were these scurs/horn from? All from the CLOV group?

Au: This information has been added.

L112: Seems like the scurs that developed from 6 to 16 months associated with CRYO were because of incomplete scur/horn removal at 6 months, and not a result of the initial CRYO treatment. Please clarify and/or discuss this for the reader.

Au: Yes, these results were not clear. We have revised this section and only scur/horn growth related to the original treatment has been reported on.

L134 (figure 1): Image (a) is a picture of a heifer with a horn that was not disbudded? And the point of the figure is to show that figures a and b look very similar?

Au: Yes, the figure was included to illustrate that a normal horn and a clove oil scur were similar in appearance. This is now clarified in the figure legend and text.

L177: Please be clear and consistent when discussing results whether referring to 6 or 16 month results.

Au: Reference to the results of the 6 month study have been removed from the results section and comparison between the 6 and 16 month results has been made more clear in the discussion.

L194: Please provide reference for this statement: it is common practice (in what country) to cautery disbud calves at 5w of age.

Au: A reference has been added.

L201: Please compare your results (related to CLOV) to those reported elsewhere in the literature, and discuss discrepancies and/or why you think those discrepancies may exist.

Au: The results of the clove oil treatment have now been compared with the results reported in the literature and discussion has been added in regard to the discrepancy among results.

L224: Conclusions: Please be consistent with use of horn vs scur.

Au: We have replaced the term horn growth with scur/horn growth to make is more clear that we are talking about both.

Round 2

Reviewer 2 Report

Our comments were not put into praxis at all, conclusion and titel are not acceptable tue to results of this study.